# Comparison of Weighted/Unweighted and Interpolated Grid Data at Regional and Global Scales

**Rui Wei** [1,2], **Yuxin Li** [1,2], **Jun Yin** [1,2,*] **and Xieyao Ma** [1,2]

1    Key Laboratory of Hydrometeorological Disaster Mechanism and Warning of Ministry of Water Resources/Collaborative Innovation Center on Forecast and Evaluation of Meteorological Disasters, Nanjing University of Information Science and Technology, Nanjing 210044, China
2    School of Hydrology and Water Resources, Nanjing University of Information Science and Technology, Nanjing 210044, China
*    Correspondence: jun.yin@nuist.edu.cn

**Abstract:** Uniform grid data are widely used in climate science and related interdisciplinary fields. Such data usually describe the hydrometeorological states averaged over uniform latitude–longitude grids. While these data have larger grid areas in the tropics than other high-latitude regions, less attention has been paid to the areal weights of these grid data. Here, we revisited two methods available for processing these uniform grid data, including weighted sample statistics and grid interpolation. The former directly considers the grid area differences using geodetic weights; the latter converts the uniform grids to equal-area grids for conventional data analysis. When applied to global temperature and precipitation data, we found larger differences between weighted and unweighted samples and smaller differences between weighted and interpolated samples, highlighting the importance of areal weights in grid data analysis. Given the different results from various methods, we call for explicit clarification of the grid data processing methods to improve reproducibility in climate research.

**Keywords:** gridded datasets; weighted sample; weighted histogram; weighted sample distribution; climate data

## 1. Introduction

In climate science, a growing amount of data is available in grid format for assessing climate change impacts. For example, climate model outputs (e.g., Coupled Model Intercomparison Project Phase 6, CMIP6, [1]), reanalysis products (e.g., European Centre for Medium-Range Weather Forecasts, ECMWF, [2,3]), remote sensing data (e.g., Moderate-resolution Imaging Spectroradiometer, MOIDS, [4]), and interpolated observations (e.g., [5,6]) usually provide hydrometeorological variables in a uniform latitude–longitude grid (hereafter referred to as uniform grid, [7]), which has the same interval in both longitude and latitude. It is clear that the area of the grid box is latitude-dependent, which can be much larger near the tropics than near the poles. Although it is usually not explicitly addressed in the literature, when identifying areal features or comparing spatial distributions from different data sources of varying resolutions, one needs to account for the areal weight in each grid point [8]. For example, a recent article published in Nature (593, 543–547, 2021) was retracted due to an error in accounting for the areal weights in grid data. Such mistakes could influence the reproducibility and repeatability of climate studies and need to be carefully investigated.

One of the approaches to account for areal weights is to treat these uniform grid data as weighted samples. For example, geodetic weights have been accounted for when calculating Earth's long-term energy balance from Clouds and the Earth's Radiant Energy System (CERES) (e.g., [9]). The weighted statistics have been used for Reynolds decomposition [10]

and for analyzing the spatial patterns of the ecosystem [11]. Geographically weighted spatial correlation coefficients, auto-correlation, and principal components analysis have been used to quantify spatial characteristics of geochemical patterns [12]. Weighted percentiles have also been provided in statistical computer packages [13]. While these statistics are useful, the full weighted density distribution has seldom been investigated, in spite of it offering detailed features across the whole spectrum of the observations.

Another approach is to interpolate these uniform grids into equal-area grids, allowing the application of any traditional statistics. Indeed, many geographical maps are projected as equal-area grids (e.g., Goode homolosine, Behrmann, Hobo–Dyer, and Lambert [14]). While useful in various applications, these projections are seldom available in the original data products and spatial interpolation may be required (e.g., [15–17]), causing biases or errors in the subsequent analysis. In particular, equal-area isolatitudinal maps provide uniform width in latitude but varying intervals in the longitude to achieve equal-area grids [18]. Given that these grids are still uniform in latitude, converting uniform grids to these equal-area isolatitudinal grids may be more efficient as discussed later in this study.

Since both weighted samples and interpolation can be used to process the uniform grid data, it is necessary to estimate their potential differences. Towards this goal, we have compared the basic statistics (averages, quartiles, distributions) of weighted, unweighted, and interpolated samples in different regions around the world using some typical climate datasets. We started by revisiting the weighted statistics and interpolation technique. When applying these to the uniform grid data of global temperature, we found clear differences between the unweighted and weighted statistics, but smaller differences between weighted and interpolated data, highlighting the importance of accounting for the areal weights in these uniform grid data. The rest of the article is organized as follows. In Section 2, we introduce the statistics of the weighted samples along with the equal-area interpolation techniques. We then use the gridded global temperature and precipitation data to compare the statistics of weighted, unweighted, and interpolated samples in Section 3. The conclusions are summarized in Section 4. Code for computing weighted sample distribution and interpolation of equal-area grids is available at https://github.com/weirrui/wDistribution and https://github.com/weirrui/eainterp (accessed on 13 July 2022).

## 2. Theory

In this section, we explain the weights of the uniform grid data in an idealized case study with latitude-dependent variable. We then briefly introduce two approaches for analyzing uniform grid data: weighted statistics and equal-area grid interpolation.

### 2.1. Idealized Case Study of the Weighted and Unweighted Global Averages

Before introducing two approaches for analyzing uniform grid data, we consider an idealized case of global variable $x$, which is only dependent on its latitude $\phi$, i.e., $x(\phi)$. The geodetic weights of this variable is also a function of latitude $w(\phi) = \cos(\phi)$, whose normalized form is

$$\overline{w}(\phi) = \frac{1}{\int_{-\pi/2}^{\pi/2} \cos(\phi)d\phi} \cos(\phi) = \frac{1}{2}\cos(\phi). \tag{1}$$

The global average with consideration of geodetic weights is

$$\mu_w = \int_{-\pi/2}^{\pi/2} x(\phi)\overline{w}(\phi)d\phi = \frac{1}{2}\int_{-\pi/2}^{\pi/2} x(\phi)\cos(\phi)d\phi, \tag{2}$$

while the unweighted average is

$$\mu_u = \frac{1}{\pi}\int_{-\pi/2}^{\pi/2} x(\phi)d\phi. \tag{3}$$

If $x(\phi) = x_0$ is constant, both averages are $_0$ as well. It is evident from this idealized case study that the differences between weighted and unweighted averages directly come from the latitude-dependent feature of the variable under consideration, which is very common in climate studies. For example, if the variable follow the latitude-dependent function of $x(\phi) = \cos(\phi)$, i.e., being maximum of 1 in the equator, and minimal of 0 in north and south poles, the weighted and unweighted average are $\pi/4$ and $2/\pi$, respectively, highlighting impacts of the weights. To explore statistics besides averages, we may consider the uniform grid data as weighted samples or interpolate the data into equal-area grids as discussed below.

*2.2. Statistics of Weighted Samples*

We consider a sample from a continuous random variable, $s_i, i = 1, 2, ..., n$ with the sample size of $n$ and non-negative weights of $w_i$, which may be treated as the frequency or number of occurrences. Regarding the uniform latitude–longitude grid data, $w_i$ refers to the geodetic weights or the relative size of each grid box (e.g., cosine of the grid latitude). The mean of the weighted samples is given as (e.g., [19,20])

$$\mu_w = \frac{1}{\sum_{i=1}^{n} w_i} \sum_{i=1}^{n} w_i s_i, \tag{4}$$

and the weighted variance is

$$\sigma_w^2 = \frac{\sum_{i=1}^{n} w_i (x_i - \mu_w)^2}{\sum_{i=1}^{n} w_i}, \tag{5}$$

This variance can be adjusted by Bessel's correction to provide unbiased estimation (e.g., [21]), although these biases may be negligible in climate studies where the high-resolution modeling outputs usually have large sample sizes. To find the distribution of the weighted sample, we can divide the sample into $k$ classes, which are centered at $x_j$ with interval $\Delta x_j$. Instead of counting the number of samples in each class, we sum the corresponding weights, i.e.,

$$C_w(x_j) = \sum_{i=1}^{n} \left[ x_j - \frac{1}{2}\Delta x_j \le s_i < x_j + \frac{1}{2}\Delta x_j \right] w_i. \tag{6}$$

where $[\cdot]$ is Iverson bracket [22]. $[P]$ is 1 if $P$ is true, and 0 if $P$ is false. The relative frequency density is the sum of the weights in each class divided by the total weights and the interval of the class, i.e.,

$$g_w(x_j) = \frac{C_w(x_j)}{\Delta x_j \sum_{i=1}^{n} w_i}. \tag{7}$$

This function also refers to the weighted histogram and has been programmed in some statistics software (e.g., SAS [23]). It tends to be the probability density function (PDF), $p(x)$, for large sample sizes and class numbers

$$p_w(x) = \lim_{\substack{\Delta x \to 0 \\ n \to \infty}} g_w(x_i), \tag{8}$$

where $\Delta x$ is the largest size of the class interval, max $\Delta x_j$. Note that this weighted sample distribution should be distinguished from the weighted distribution proposed by Rao [24]. Weighted distribution is associated with non-random sampling methods and the corresponding weights are expressed as the function of observations (e.g., $w_i = f(s_i)$), whereas the distribution of the weighted sample discussed here can have different weights for observations of the same value.

With estimated $p_w(x)$, we obtain the cumulative distribution function (CDF), $P_w(x)$

$$P_w(x) = \sum_{-\infty}^{x} p_w(x')dx'. \tag{9}$$

Its inverse function can be used for evaluating the weighted percentiles, $x = P_w^{-1}(k)$. Specifically, when $x_j$ is chosen to be the unique values in the sample $s_i$, the corresponding algorithm for $P_w^{-1}(k)$ has been programmed in some software to calculate the percentiles of the weighted sample (e.g., [13]).

### 2.3. Interpolation for Equal-Area Grids

While weighted statistics should be the first choice for processing grid data since it is the accurate way to consider the weights, we may need to convert uniform grids to equal-area grids if more complicated statistics are required. To minimize biases introduced by interpolation, it is reasonable to set the new grids close to the original ones. In this regard, we may simply keep the meridional component but change the zonal interval to have equal-area grids, which is referred to as the equal-area isolatitudinal grid [18] and has been used in many data sets, such as the International Satellite Cloud Climatology Project [25]. An illustrative example for such a grid conversion is provided in Figure 1. We keep the old grid structure in the middle of the study domain (i.e., no interpolation) and use the same latitude interval for the new grid. To keep the same grid area, the number of the new grid in each latitude interval $N_{n,i}$ should be inversely proportional to the total area in the corresponding latitude $\phi_i$,

$$\frac{N_{n,i}}{\cos \phi_i} = \frac{N_o}{\cos \phi_c} \tag{10}$$

where $N_o$ is the grid number in each interval of latitude of the old grid, and $\phi_c$ is the latitude at the center of the study domain. Since the number of grids needs to be an integer, Equation (10) can be approximated as

$$N_{n,i} \approx \lfloor \frac{N_o \cos \phi_i}{\cos \phi_c} \rceil. \tag{11}$$

where $\lfloor \rceil$ refers to the function of the nearest integer. For global study, the center of the domain is the equator and Equation (11) reduces to $N_{n,i} \approx \lfloor N_o \cos \phi_i \rceil$.

(a)  uniform grids

$N_0$ grids in each latitude interval

centered at $\phi_c$

(b)  equal-area grids

$N_{n,i}$ grids in each $\phi_i$ interval

centered at $\phi_c$

**Figure 1.** An illustrative example of converting (**a**) uniform latitude–longitude grids to (**b**) equal-area isolatitudinal grids. There are constant $N_0$ grids in each latitude interval in (**a**) but varying $N_{n,i}$ grids in (**b**).

## 3. Applications

In this section, using the methods discussed above, we compare weighted and unweighted statistics (means, quartiles, distributions) in different regions of the world using some typical global datasets. We also compare the distributions of weighted and interpolated samples.

### 3.1. Statistics from Weighted and Unweighted Samples

Since not all climate studies have explicitly clarified how to process the uniform latitude-longitude grids, it is possible that these datasets have been accidentally treated as regular samples and induced certain biases in the corresponding statistical analysis (e.g., Nature vol. 593, 543–547, 2021). To estimate these biases, we first compare the basic statistics of weighted and unweighted (i.e., regular) samples using the uniform grid temperature data from Climatic Research Unit (CRU TS v. 4.06, crudata.uea.ac.uk/cru/data/hrg/, [26] (accessed on 13 July 2022)) with grid resolution of $0.5 \times 0.5°$. We perform statistical analysis not only for the whole land area (excluding Antarctica) but also in three large countries (China, United States, and Canada) and three different zonal regions (0–10, 0–20, 0–40, 0–60 °N), covering the low-, mid-, and high-latitudes and with different latitudinal widths. We compared the temperature time series averaged over the study domain with and without consideration of the areal differences of the grid boxes; we also compared the corresponding quartiles and the spatial distributions of temperature in the year 2021.

We calculated the time series of arithmetic averages (i.e., unweighted, $\mu_u$) and averages with geodetic weights (i.e., weighted, $\mu_w$ with $w_i = \cos(\phi_i)$) of the near-surface air temperature over the four study domains (see Figure 2). We found that the gaps between $\mu_w$ and $\mu_u$ can be as high as 5 K across the globe. At region scales, $\mu_w$ and $\mu_u$ are very close in China but clearly different in the US and Canada with higher $\mu_w$ than $\mu_u$. The differences in the United States are mainly due to the presence of high-latitude Alaska. If the study domain is reduced to the continental United States, the differences between $\mu_w$ and $\mu_u$ are similar to those in China. Canada has wider latitude range than China and the United States, and the corresponding differences between $\mu_w$ and $\mu_u$ are even greater. This is corroborated by the comparisons among various zonal regions, where the largest latitude range (0–60 °N) has the biggest differences between $\mu_w$ and $\mu_u$. Table 1 reported the mean absolute errors (MAE) and root mean square errors (RMSE) of these differences. Similar patterns can be found for the time series of precipitation (see Figure 3).

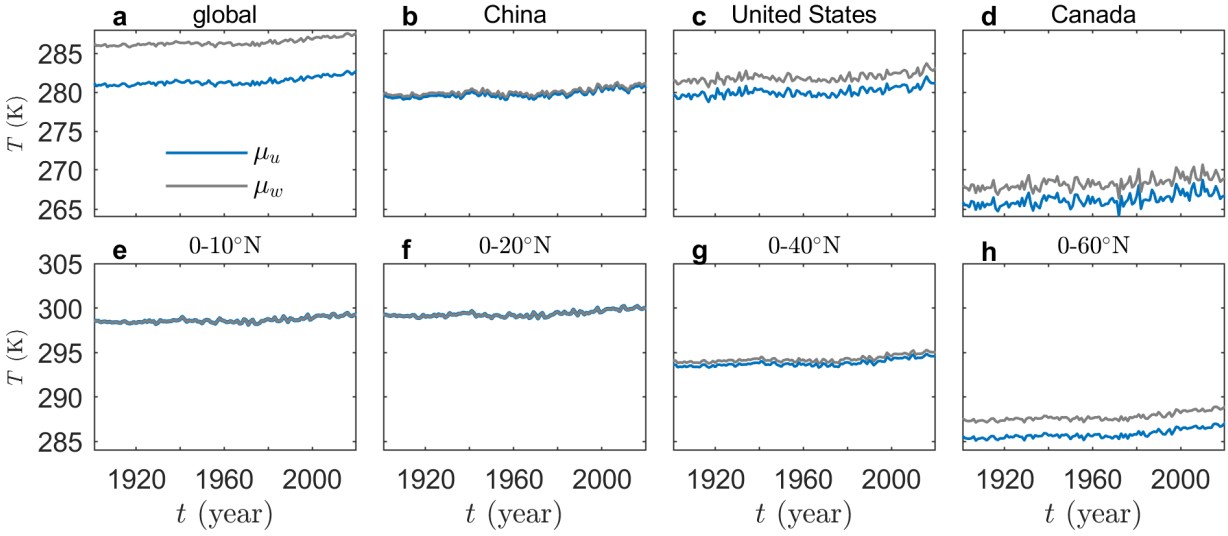

**Figure 2.** Comparison of weighted and unweighted averages ($\mu_w$ and $\mu_u$) of temperature time series (**a**) around the globe, across (**b**) China, (**c**) the United States, (**d**) Canada (**e**) 0–10 °N, (**f**) 0–20 °N, (**g**) 0–40 °N, and (**h**) 0–60 °N using CRU data.

**Table 1.** MAE and RMSE among the weighted and unweighted temperature time series in 1901–2021 using CRU data.

| Region | Global | China | United States | Canada | 0–10 °N | 0–20 °N | 0–40 °N | 0–60 °N |
|--------|--------|-------|---------------|--------|---------|---------|---------|---------|
| MAE | 5.0202 | 0.3491 | 1.8592 | 2.1612 | 0.0013 | 0.0096 | 0.4656 | 1.9577 |
| RMSE | 5.0207 | 0.3498 | 1.8621 | 2.1623 | 0.0013 | 0.0096 | 0.4657 | 1.9582 |

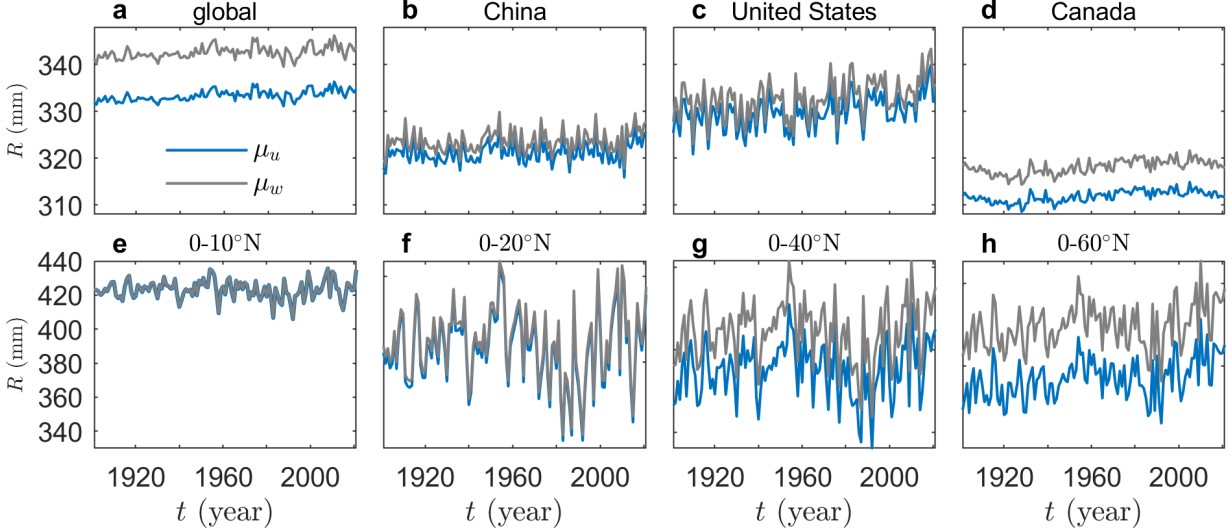

**Figure 3.** As in Figure 2 but for the precipitation.

While the temperature time series seem in parallel for both $\mu_u$ and $\mu_w$ from Figure 2, certain differences can still be identified. As shown in Figure 4, the difference is about $-5.2$ K in 1900 but decreases to $-4.9$ K in 2020, consistent with global warming trends. The faster warming rate from unweighted averages is due to the over-representation of the arctic regions, which tend to have faster warming rates [27,28].

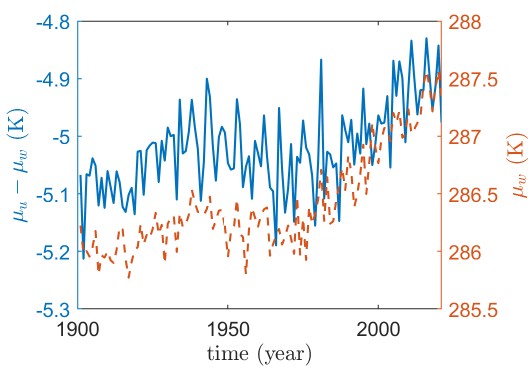

**Figure 4.** Differences of weighted and unweighted averages of global temperature ($\mu_u - \mu_w$). Time series of global mean temperature $\mu_w$ is also shown for reference.

The weighted and unweighted first quartile, denoted respectively as $Q_{1,w}$ and $Q_{1,u}$, are reported in Figure 5. Similar to $\mu_w$ and $\mu_u$, the differences between $Q_{1,w}$ and $Q_{1,u}$ can be as high as 7 K at the global scale, whereas at regional scales the differences are very close in China but greater in the United States and Canada. Overall, $Q_{1,w}$ and $Q_{1,u}$ differences are greater than those between $\mu_w$ and $\mu$, illustrating that the weighting impacts can be greater for extreme values. Similar patterns can be found for the inter-quartile range (IQR) defined as the differences between the first and third quartiles (see Figure 6). These extremes were further explored by comparing the weighted and unweighted histograms below.

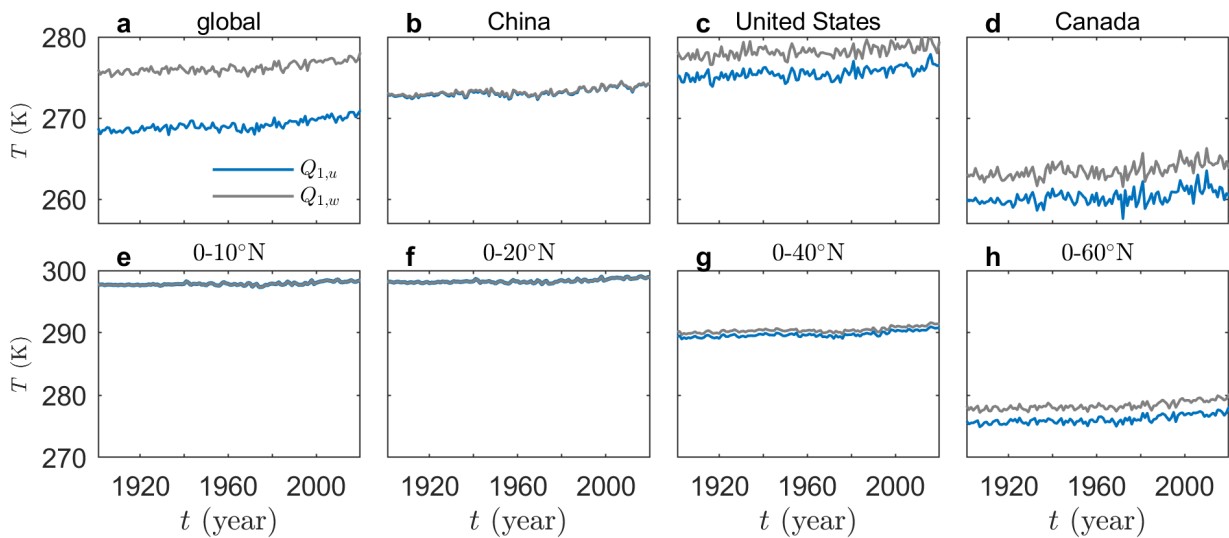

**Figure 5.** As in Figure 2 but for the first quartile with and without consideration of geodetic weights ($Q_{1,w}$ and $Q_{1,u}$).

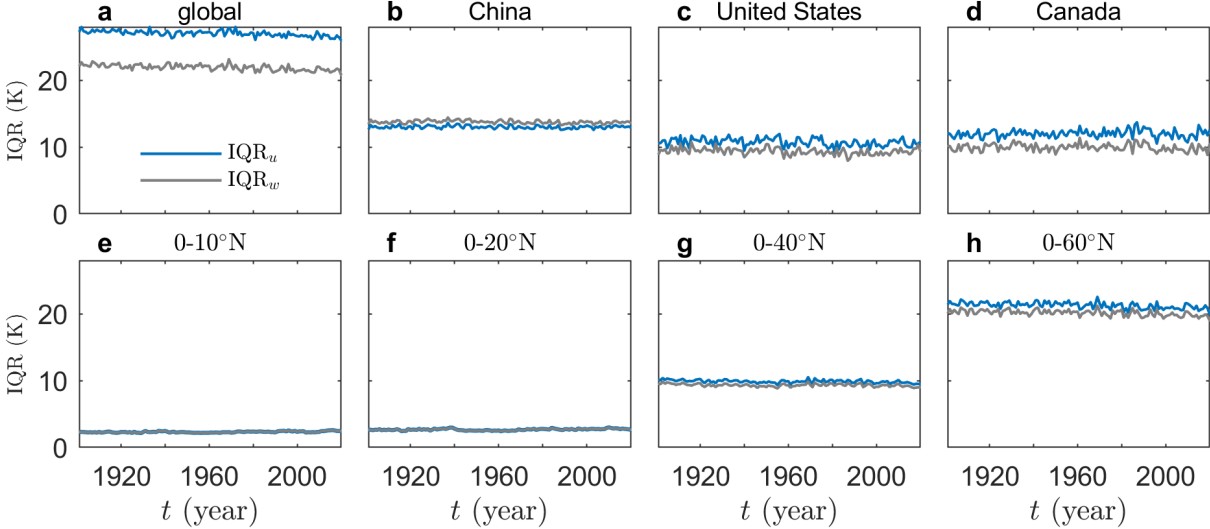

**Figure 6.** As in Figure 2 but for the interquartile range (IQR) with and without consideration of geodetic weights (IQR$_w$ and IQR$_u$).

The probability density functions in the year of 2021 from grid data with and without consideration of geodetic weights (i.e., $p_w(T)$ and $p_u(T)$) are presented in Figure 7. At the global scale, the differences are large in the left tails (low-temperature regime), whereas they are smaller in the right tails (high-temperature regime). At a regional scale, $p_w(T)$ and $p_u(T)$ are almost identical in China but greater in the US and Canada. In general, the differences are greater for wider latitude ranges (see Figure 7e–h). These distributions explain that the differences between averages and quartiles in Figures 2 and 5 come from the over-representation of low temperature in the unweighted samples. With denser grids in high-latitude regions in the uniform grid data, unweighted averages and quartiles are certainly lower and cannot represent the real observations.

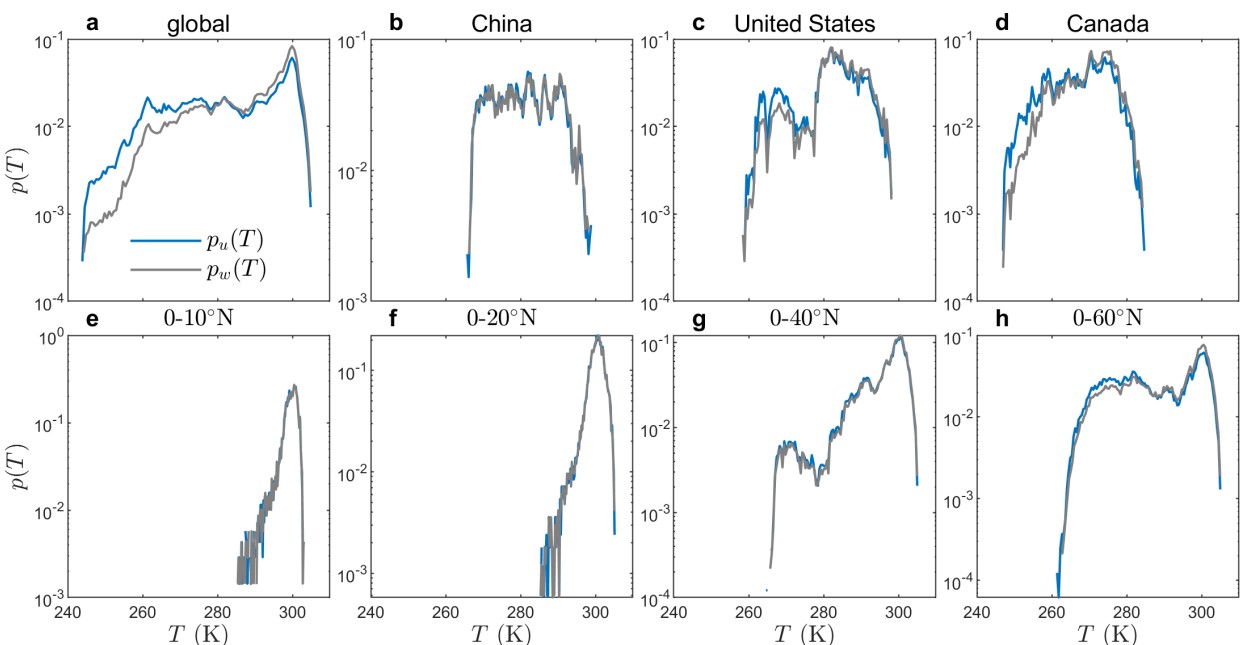

**Figure 7.** As in Figure 2 but for temperature distributions with and without consideration of geodetic weights ($p_w(T)$ and $p_u(T)$).

### 3.2. Distributions from Weighted and Interpolated Samples

To quantify the biases induced by the interpolation, we compared the distributions from weighted and interpolated samples. We applied the grid conversion in Section 2.3 to the CRU data by setting the center of the domain as the equator and using bilinear interpolation. It should be noted that, aside from this bilinear interpolation, equal-area grids provided in this study can be directly applied with any interpolation technique, which may be more suitable for capturing the specific spatial heterogeneity (e.g., elevation-dependent spatial information).

As reported in Figure 8, these two temperature distributions, $p_w(T)$ and $p_i(T)$, show smaller differences at the global scale but slightly larger differences at regional scales, suggesting our equal-area isolatitudinal grid interpolation could be a valid method for processing climate data.

The biases induced by interpolation may also depend on the spatial variations of the variable. To address this point, we also investigate the distributions of precipitation, another typical hydrometeorological variable. As reported in Fig. 6, the global distributions of weighted and interpolated precipitation, $p_w(R)$ and $p_i(R)$, are almost identical for $R \leq 200$ mm/month but slightly different for $R > 200$ mm/month. It should be noted that the distributions are on a logarithmic scale. Similar patterns were also observed at regional scales with greater differences in precipitation distributions for the extreme values, suggesting greater interpolation biases over the wetter regions.

The resolutions of the data determine the amount of information available for interpolation and thus influence the accuracy of the derived data. For this reason, we also investigated the temperature data from HadCRUT, which has a much coarser resolution of $5 \times 5°$ in both latitude and longitude (crudata.uea.ac.uk/cru/data/temperature, [29,30] (accessed on July 2022)). As discussed later, $p_w(T)$ and $p_i(T)$ have smaller differences at global scale. When compared with the CRU data (i.e., Figures 9 and 10), the differences between $p_w(T)$ and $p_i(T)$ are more evident for data with coarse resolutions, which provide less accurate information for spatial interpolation and consequently result in greater biases.

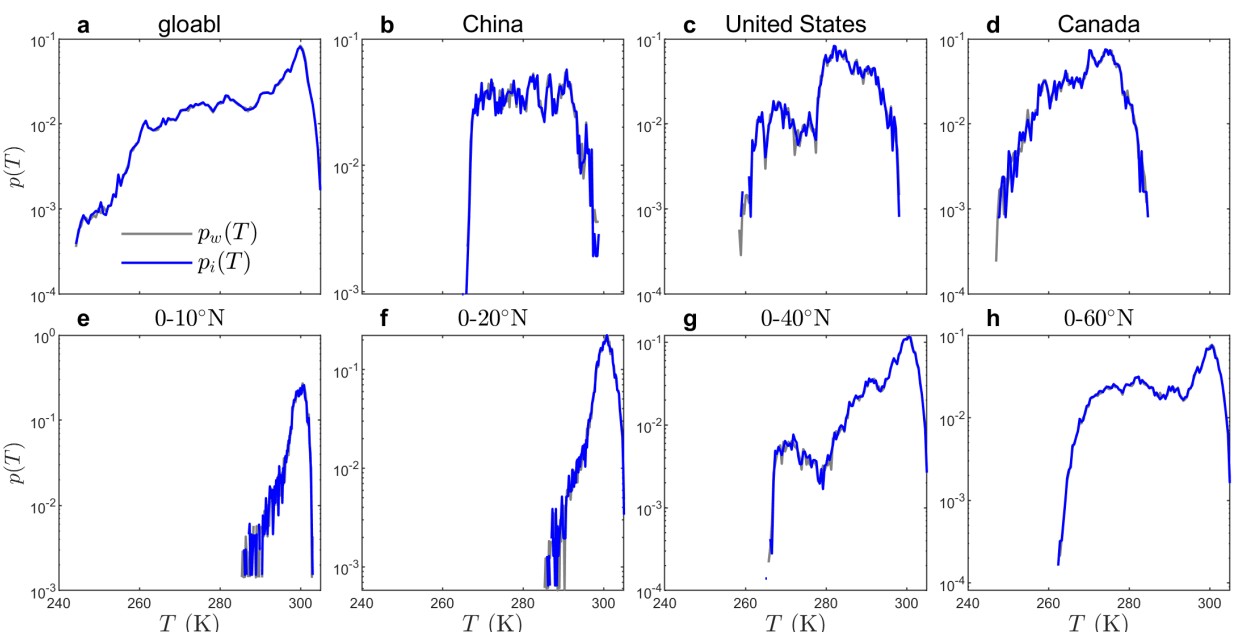

**Figure 8.** Comparison of the temperature distributions from weighted and interpolated samples in 2021 (**a**) over the globe, across (**b**) China, (**c**) the United States, and (**d**) Canada (**e**) 0–10 °N, (**f**) 0–20 °N, (**g**) 0–40 °N, (**h**) 0–60 °N using CRU data.

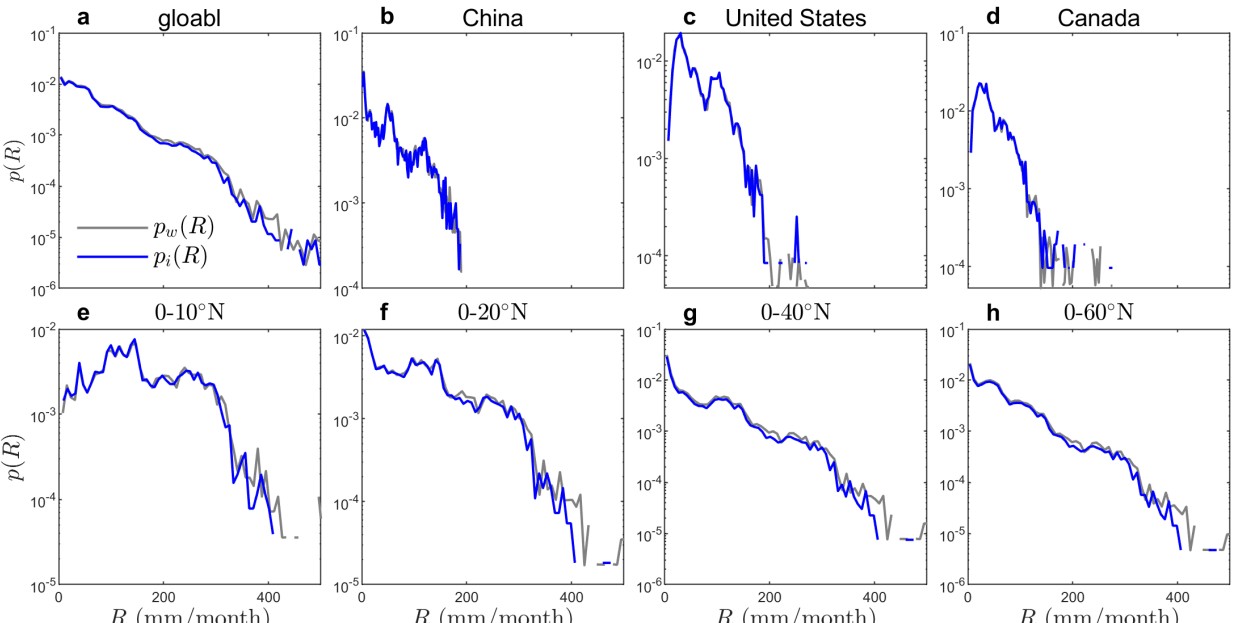

**Figure 9.** As in Figure 8 but for precipitation distributions.

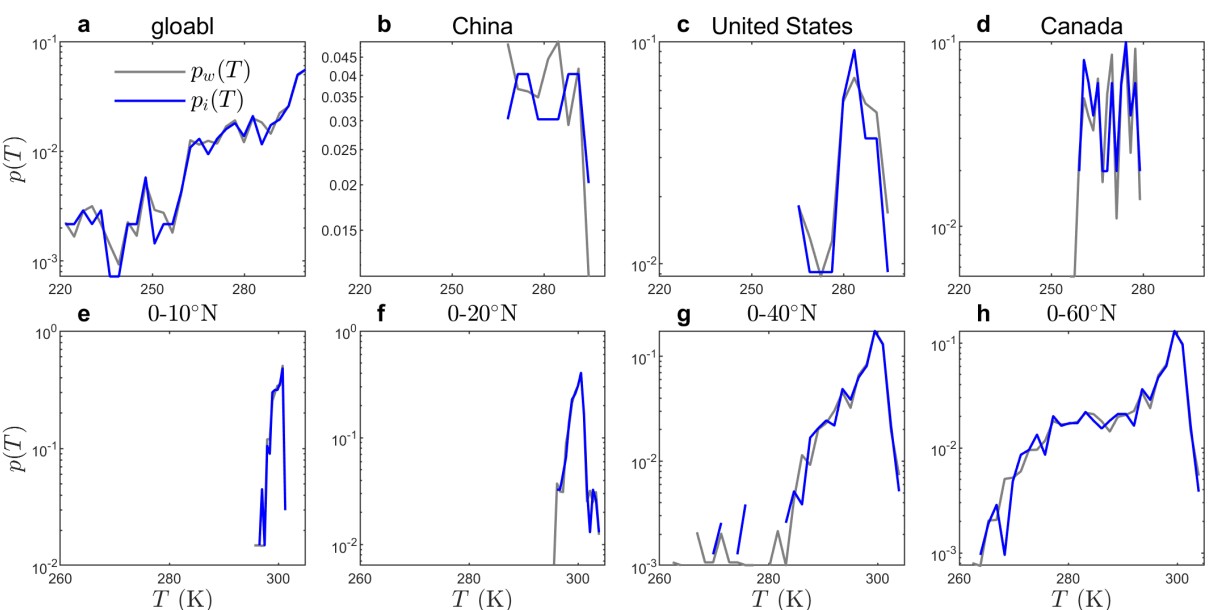

**Figure 10.** As in Figure 8 but for HadTEMSST data.

## 4. Conclusions

In this study, we revisited the methods available for processing uniform latitude–longitude grid data, which have uneven grid areas across different regions of the world. We explained weighted sample statistics and provided mathematical expressions for calculating the weighted sample distribution. We also introduced the interpolation approach to converting uniform grids to equal-area grids by setting the new grids close to the old ones. These two methods were then applied to global temperature and precipitation data to compare statistics from both weighted and interpolated samples. The results not only show larger differences between weighted and unweighted samples and smaller ones between weighted and interpolated samples, but also demonstrate that these differences depend on the locations, variables, and resolutions of the data.

These findings have significant implications for the future applications of grid data extensively used in climate studies. The large differences between weighted and unweighted samples suggest that we should consider the areal weights when processing the uniform grid data. Biases or errors can be large in study domains with large latitude ranges and/or in high-latitude regions. When conventional statistics are not available for weighted samples, converting to equal-area grids is a promising method, which can be accurate as long as the data are relatively smooth and in high spatial resolutions. Given the clear differences among various approaches, we call for explicit clarification of the data processing methods, thus improving the reproducibility and replicability of future climate studies.

**Author Contributions:** Conceptualization, R.W., Y.L., J.Y. and X.M.; methodology, R.W., Y.L. and J.Y.; software, R.W.; validation, R.W.; formal analysis, R.W., Y.L. and J.Y.; investigation, R.W., Y.L. and J.Y.; resources, J.Y.; data curation, R.W.; writing—original draft preparation, R.W.; writing—review and editing, R.W., Y.L. and J.Y.; visualization, R.W.; supervision, J.Y.; project administration, J.Y.; funding acquisition, J.Y. and X.M. All authors have read and agreed to the published version of the manuscript.

**Funding:** This study is supported by the Natural Science Foundation of Jiangsu Province (BK20221343), the National Natural Science Foundation of China (41877158, 51739009), and NUIST startup funding (1441052001003).

**Data Availability Statement:** CRU TS v. 4.06 temperature and precipitation data can be download from crudata.uea.ac.uk/cru/data/hrg/ (accessed on 13 July 2022) and HadCRUT temperature data are available at crudata.uea.ac.uk/cru/data/temperature (accessed on 13 July 2022).

**Acknowledgments:** We acknowledge support from NUIST's supercomputing center. The comments and useful criticisms of four anonymous reviewers are gratefully acknowledged.

**Conflicts of Interest:** The authors declare no conflict of interest.

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
