# Peer review of "Comparison of Weighted/Unweighted and Interpolated Grid Data at Regional and Global Scales"

_atmosphere, doi:10.3390/atmos13122071_

Round 1

Reviewer 1 Report

Spatial averaging is one of the most commonly used mathematical methods in Earth sciences. The spatial grid homogeneity of the data products overcome the problem of uneven spatial distribution of the observation, thus, is widely used in climate change related studies. However, the spherical nature of the Earth makes the latitude and longitude grid density gradually increase from the equator to the poles, which leads to overweighting of the data products at high latitudes. Nevertheless, this issue is often overlooked, and an article recently published in nature (Nature vol. 593, 543–547 (2021)) had been retracted because of this flaw. To address this issue, this paper compares the effects of three different averaging methods on the calculation results and points out the necessity of area weight averaging in large scale studies. The study is methodologically sound and the results are credible. The article is generally well organized and presented. I think the conclusions of the study could be further solidified in the following aspects.

General comments:

The authors analyzed the differences between the results of different averaging methods using the temperature variable as an example and found that the found that the area-weighted averaging method and the arithmetic averaging method can differ by up to 5 degrees globally. But for the other variables, it is difficult to grasp the magnitude of the impact of this difference in area weights. I suggest that the authors could, in Section 2, mathematically set the background value of the variable to 1 or values with some gradient variation, and then compare the differences in the mean values obtained by the different methods, which would facilitate the understanding of the conservative estimation of the area weight effect.

In addition, this paper only analyzes the differences in the mean values, so are the trends in the time series obtained by different averaging methods consistent? Taking the temperature variable as an example, recent studies found that the rate of warming in the Arctic is four times higher than the global average. So does this Arctic amplification effect affect the estimated global average warming rate when different averaging methods are used?

It is well known that topography is usually a key determinant of the spatial heterogeneity of atmospheric variables. Did the authors consider the elevation-dependent pattern of temperature or precipitation when calculating the average of area weights using spatial interpolation?

Specific Comments:

Lines 120-122: No need to give a conclusion here.

Line 165: bilinear interpolation

Lines 216-218: Please indicate the source of the precipitation data used in this paper.

Author Response

Dear Reviewer 1

We would like to thank you for the careful and constructive assessment of our manuscript. Following your suggestions and comments, we have thoroughly revised the manuscript as detailed in the attachment.

Your sincerely,

Jun Yin

Reviewer 2 Report

This study revisited two methods available (geodetic weights and interpolation) for processing uniform grid data. A larger difference exists for global mean temperature and precipitation between the weighted and unweighted schemes, especially for the high latitude domain. This is common sense. I don’t think this can be highlighted in the abstract. Other more important findings should be summarized in the abstract. In all, this work can give some useful information for climate research. I recommend accepting it after a minor revision.

Specific comments:

1. The reason for selecting those three national regions in your work (China, United States, and Canada) should give some explanation. I recommend selecting a tropic region in your research to highlight the little difference between weighted and unweighted in the low-latitude domain.

2. Authors said that the little difference between the weighted and interpolated. Could you give us some suggestions that the condition to choose the weighted or interpolated in the climate research? 

3. It is interesting to show the range between the first and third quartiles for weighted and unweighted schemes. 

Author Response

Dear Reviewer 2

Thank you for the careful and constructive assessment of our manuscript. We have thoroughly revised the manuscript as detailed in the attached file.

Yours sincerely,

Jun Yin

Reviewer 3 Report

This study compared the basic statistics of weighted and unweighted samples using grid temperature data from CRU. The authors found larger differences between weighted and unweighted samples and smaller differences between weighted and interpolated samples. This work is of practical significance to the use of grid data. However, the evaluation is very basic and more detailed analysis are required. The global pattern of the grid data and an evaluation with MAE and RMSE are suggested. Furthermore, there is no discussion on the mechanism behind the differences between weighted and unweighted samples. A deep discussion is essential for this study.

Author Response

Dear Reviewer 3

We would like to thank you for the careful and constructive assessment of our manuscript. Following your suggestions and comments, we have thoroughly revised the manuscript as detailed in the attachment.

Regards,

Jun Yin

Reviewer 4 Report

The authors examined effects of weighting on an assessment of gridded climate data. The structure of introduction is clear. The description of methodology is specific. The topic of the study falls into the scope of the journal. However, results should be given in a more clear and specific way.

Major concerns:

The authors mentioned the area of grid box is latitude-dependent, but only studied three countries, China, United States and Canada. It is better to give a result of difference between weighted and unweighted grid data along the gradient of latitude with an interval of 10° or 20° at the global scale. In addition, the authors could select two sites (one of low latitude and one of high latitude) in three countries respectively, and examined weighting effects on gridded climate data. Since the latitudinal range is relatively large for three countries. The results clearly showed that weighting calculations differently impact temperature data and precipitation data. Results of weighting effects on time-series precipitation data were not found. The related discussion should be given in a detailed way.

Specific concerns:

The scale of y-axis is not consistent for some figures, like figure 5 and 7.

In figure 7, it is very confused that China occurred three times (b, c and d). Please check the errors throughout the manuscript.

Author Response

Dear Reviewer 4

We would like to thank you for the careful and constructive assessment of our manuscript. Following your suggestions and comments, we have thoroughly revised the manuscript as detailed in the attachment.

Yours sincerely

Jun Yin

Round 2

Reviewer 3 Report

The authors have made great improvements on the manuscript. I believe the current version can be accept. 

Reviewer 4 Report

The revised manuscript improved a lot and could be accepted in the present form